# The Genetics and Evolution of Human Pigmentation

**DOI:** 10.3390/biology14081026

**Published:** 2025-08-10

**Authors:** Dorra Guermazi, Elie Saliba

**Affiliations:** 1Department of Dermatology, Warren Alpert Medical School of Brown University, Providence, RI 02903, USA; elie_saliba@brown.edu; 2Department of Dermatology, Gilbert and Rose-Marie Chagoury School of Medicine, Lebanese American University, Beirut 13-5053, Lebanon

**Keywords:** skin pigmentation, evolution, UV radiation, genetics, natural selection, MC1R, SLC24A5, TYR, OCA2, convergent evolution

## Abstract

Skin color varies dramatically among people across the world, but the reasons behind this variation are complex and rooted in our evolution. This article explains how our genes, sunlight exposure, and geography have shaped human skin pigmentation over thousands of years. We describe how different populations have adapted to their environments—for example, darker skin protects people living near the equator from strong sunlight, while lighter skin helps people in colder regions obtain enough vitamin D from the sun. We also show how similar skin tones developed in different parts of the world through different genetic paths. Understanding this evolution helps scientists and doctors better address health concerns like vitamin-D deficiency and skin cancer. It also highlights the importance of including people of all skin types in medical research and education. This study combines insights from genetics, history, anthropology, and medicine to tell the story of skin color and why it matters for health and society.

## 1. Introduction

Skin pigmentation is one of the most visible examples of human adaptation to environmental conditions. Its variation is strongly shaped by geographic differences in ultraviolet (UV) radiation, with early theories emphasizing melanin’s protective role against sunlight. Over time, research has evolved from basic observational studies to cutting-edge genome-wide association studies, which highlight the complex genetic architecture and evolutionary forces that influence pigmentation. Key recent contributions include the analysis of pigmentation-associated loci in diverse African populations [1,2], findings from the Human Genome Diversity Project [3], and evidence of recent selection from ancient genomes spanning Europe, Central Asia, and the Americas [4,5]. This expanding body of work not only deepens our understanding of human evolutionary history but also carries important implications for health, particularly regarding disease susceptibility, vitamin-D synthesis, and responses to UV exposure in diverse populations.

## 2. Genetic Architecture of Skin Pigmentation

### 2.1. Melanin Biosynthesis and Pigment Cell Biology

Melanin, the primary determinant of human skin color, is synthesized by melanocytes—specialized pigment-producing cells located in the basal layer of the epidermis. These melanocytes produce two main types of melanin: eumelanin, which is brown to black, and pheomelanin, which is red to yellow [6]. The balance between these two types largely determines the overall skin tone, with higher eumelanin levels contributing to darker pigmentation and increased photoprotection.

Melanin synthesis occurs through a complex biochemical pathway known as melanogenesis. The key enzyme driving this process is tyrosinase (encoded by the TYR gene), which catalyzes the oxidation of the amino acid tyrosine into DOPA and then to dopaquinone—critical early steps in melanin production. Additional enzymes, including tyrosinase-related protein 1 (TYRP1) and dopachrome tautomerase (DCT), further regulate the quantity, quality, and type of melanin produced by modulating downstream steps in the pathway [7]. Together, these enzymatic activities influence melanin composition and ultimately affect pigmentation phenotypes. Figure 1 illustrates the melanogenesis pathway in detail.

This diagram outlines the biochemical steps of melanogenesis—the process through which melanin is synthesized in melanocytes. The pathway begins with tyrosine (purple), which is enzymatically converted into DOPA and then dopaquinone (both shown in orange), catalyzed by tyrosinase. At dopaquinone, the pathway diverges into two branches: the eumelanin pathway (left, in orange), which leads to the production of brown–black eumelanin, and the pheomelanin pathway (right, in blue), resulting in red–yellow pheomelanin. The balance between these pathways is regulated by genetic factors (e.g., MC1R variants), hormonal signals (e.g., α-MSH), and environmental influences such as UV radiation. The type of melanin produced determines skin color and photoprotective capacity, with eumelanin offering greater UV protection compared to pheomelanin.

### 2.2. Major Genetic Determinants

Human pigmentation is regulated by a complex network of genes that coordinate melanin biosynthesis, melanosome formation, and the transport and distribution of melanin within the skin. These genetic components interact within melanocytes to control both the amount and type of melanin produced [6]. Among the most extensively studied is the melanocortin 1 receptor gene (MC1R), which plays a pivotal role in determining the shift between eumelanin (brown–black) and pheomelanin (red–yellow) production. The activation of MC1R by its ligand, α-melanocyte-stimulating hormone (α-MSH), stimulates cyclic AMP (cAMP) signaling, promoting eumelanin synthesis. Conversely, loss-of-function variants in MC1R disrupt this pathway, favoring increased pheomelanin production [8]. These variants are particularly prevalent in European populations and are strongly associated with lighter skin, red hair, increased freckling, and heightened susceptibility to UV-induced skin damage and melanoma. This signaling pathway has been well-characterized across diverse ethnic groups, including admixture mapping studies in African-American and Latino cohorts [9,10].

Another critical gene influencing pigmentation variation between African and non-African populations is SLC24A5, which encodes a potassium-dependent sodium/calcium exchanger located on the membrane of melanosomes—the specialized organelles within melanocytes where melanin is synthesized and stored [11]. The A111T polymorphism (alanine to threonine substitution at codon 111) in SLC24A5 is nearly fixed in European populations and is believed to alter melanosome pH, thereby affecting the enzymatic efficiency of melanogenesis. This variant exhibits strong signatures of positive selection, underscoring its adaptive advantage in low-UV environments where lighter skin facilitates vitamin-D synthesis.

OCA2, a gene initially characterized for its involvement in oculocutaneous albinism type 2, also plays a crucial role in normal pigmentation. It influences melanosomal pH and regulates the transport of tyrosine, a key melanin precursor [12]. The expression of OCA2 is partially controlled by a well-characterized single nucleotide polymorphism (SNP) located within an intronic region of the neighboring gene HERC2. This regulatory SNP modulates OCA2 expression levels, affecting pigmentation phenotypes. Notably, a specific allele of this SNP is associated with blue eye color and reduced melanin content in the skin and hair, and it is predominantly found in European populations [13]. These key genes are summarized in Table 1.

### 2.3. Additional Genes and Polygenic Interactions

Beyond the major genes described above, a constellation of additional genes significantly contributes to the broad spectrum of pigmentation variation observed across human populations. For example, KITLG (KIT ligand) is essential for melanocyte development, survival, and migration. Variants in KITLG have been associated with lighter skin pigmentation in Eurasian populations and are thought to influence melanocyte density or melanin production during embryogenesis [14].

The ASIP gene (agouti signaling protein) encodes an antagonist of MC1R and can reduce eumelanin synthesis by competitively inhibiting α-MSH binding [15]. ASIP also plays a role in pigmentation patterning, contributing to regional differences in melanin distribution across the body.

SLC45A2 encodes a transporter involved in regulating melanosome pH and is subject to strong selective pressure in non-African populations [16]. Polymorphisms in SLC45A2 are associated with lighter skin, hair, and eye color, and mutations in this gene have been implicated in pigmentation disorders.

Collectively, these genes form a coordinated network that governs not only the biochemical pathways of pigmentation but also reflects the evolutionary adaptations of human populations to diverse environmental pressures. Their combined genetic variation underlies the polygenic architecture of pigmentation traits, influenced by additive effects as well as gene–gene interactions (epistasis). The allele frequencies and phenotypic expressions of these genes illustrate how natural selection has shaped human biology in response to factors such as UV radiation exposure, dietary vitamin-D requirements, and possibly social or sexual selection pressures.

Understanding this complex genetic architecture is crucial for reconstructing human evolutionary history and for interpreting the clinical implications of pigmentation-related diseases and conditions in contemporary populations.

## 3. Gene Interactions and Epigenetic Regulation

### 3.1. Epistasis and Gene Interactions

Although individual gene variants are important determinants of pigmentation, the trait is fundamentally polygenic, influenced by multiple genes that often interact in complex ways. These gene–gene interactions, known as epistatic effects, can substantially modify pigmentation phenotypes beyond the simple additive effects of single variants.

For example, the phenotypic impact of MC1R variants—which shift melanin production toward pheomelanin when inactivated—can be altered by the expression of the ASIP gene [17]. ASIP encodes an antagonist of MC1R, and elevated ASIP expression can reduce eumelanin synthesis, even in individuals with a fully functional MC1R [18]. This illustrates how regulatory crosstalk between genes modulates pigmentation outcomes through epistatic interactions. Recent epistatic models have demonstrated significant interactions between MC1R and OCA2 in modulating not just pigmentation but freckling patterns and sunburn sensitivity [19,20].

### 3.2. Epigenetic Regulation

In addition to genetic interactions, epigenetic mechanisms play a significant role in the regulation of pigmentation-related genes. DNA methylation, histone modifications, and non-coding RNAs can alter gene expression without changing the underlying DNA sequence. These epigenetic marks influence chromatin accessibility and transcriptional activity in melanocytes—the specialized cells that produce melanin. For example, increased methylation of promoter regions in genes like TYR or MITF can downregulate their expression, leading to decreased melanin synthesis.

Importantly, environmental factors such as ultraviolet (UV) radiation can dynamically influence epigenetic patterns. Chronic UV exposure not only increases melanin production as an adaptive response but can also lead to persistent epigenetic changes that affect pigmentation long-term [14,21]. Studies have shown that UV exposure can trigger global hypomethylation or specific histone acetylation in melanocytes, resulting in altered expression of pigmentation genes. This dynamic interplay between the genome and environment underscores the plasticity of pigmentation traits and the importance of epigenetic regulation as a mediator of environmental responsiveness. Genome-wide methylation studies have further linked pigmentation plasticity to environmental exposures beyond UV, including pollutants and diet [22,23].

### 3.3. Population-Level Allelic Variation

Population genetics studies reveal considerable diversity in pigmentation-associated alleles across global populations, shaped by evolutionary forces such as natural selection, genetic drift, and migration. In sub-Saharan African populations, there is a high prevalence of ancestral alleles that favor eumelanin production, which provides robust protection against intense UV radiation. These alleles contribute to darker skin pigmentation, reducing the risk of UV-induced DNA damage and preventing the photodegradation of folate—a critical nutrient in fetal development and cellular division.

In contrast, populations in Europe and East Asia have independently evolved sets of genetic variants associated with lighter skin pigmentation. For example, the A111T variant in SLC24A5 and the L374F variant in SLC45A2 are nearly fixed in European populations, whereas East Asians commonly carry different variants, such as those in OCA2 and DDB1/TMEM138 [24]. This phenomenon, known as convergent evolution, reflects how similar selective pressures—namely reduced UV exposure—can lead to the emergence of lighter skin via distinct genetic pathways. Such parallel adaptations highlight the flexibility of the human genome in responding to environmental demands.

The genetic architecture of pigmentation across populations is thus not only a record of human migration and demographic history but also a reflection of localized evolutionary pressures acting on ancestral groups.

### 3.4. Adaptive Evolution of Pigmentation Traits

The global distribution of human skin pigmentation is a vivid illustration of how natural selection shapes visible human traits in response to environmental variation—most notably, ultraviolet (UV) radiation. In equatorial regions, where UV exposure is intense and consistent throughout the year, darker skin pigmentation has been strongly favored due to its photoprotective properties. The high eumelanin content in dark skin acts as a natural sunscreen, effectively absorbing UV radiation and protecting against both DNA damage and the photodegradation of folate, a critical nutrient for fetal neural tube development and cellular function [14,21]. Alleles that support this pigmentation pattern—such as conserved functional variants in MC1R, TYR, and OCA2—remain prevalent in populations from these regions, reflecting strong and stable selective pressures over tens of thousands of years [6,25].

In contrast, populations residing at higher latitudes faced a different evolutionary challenge: obtaining sufficient vitamin D from limited UV exposure. Under low-UV conditions, darker skin becomes a disadvantage, as high melanin levels reduce the skin’s ability to synthesize vitamin D3. To adapt, lighter skin pigmentation evolved to increase UV penetration and enable more efficient endogenous vitamin-D production, which is vital for calcium absorption, bone growth, and immune function [14,26]. This evolutionary shift occurred independently in different populations—a phenomenon known as convergent evolution. For instance, lighter skin in Europeans evolved primarily through variants in SLC24A5, SLC45A2, and HERC2/OCA2, whereas East-Asian populations exhibit lighter pigmentation due to entirely different sets of genetic variants, such as those in DDB1, OPRM1, and other loci [10,24]. The processes mentioned in this section are outlined in Figure 2.

This diagram illustrates how variation in ultraviolet (UV) radiation across geographic regions has driven the evolution of skin pigmentation through natural selection. UV radiation exposure (yellow) acts as a selective pressure (purple), leading to two distinct evolutionary trajectories. In equatorial regions with high UV exposure (left, orange pathway), darker skin pigmentation evolved through increased eumelanin production, offering enhanced photoprotection and preservation of folate—a critical nutrient for fetal development and cellular function. This adaptation is associated with conserved variants in genes (green) such as MC1R, TYR, and OCA2. In contrast, populations living in high-latitude regions with low UV exposure (right, blue pathway) evolved lighter skin pigmentation and reduced melanin levels to facilitate more efficient vitamin-D synthesis. These adaptations arose via distinct genetic mechanisms (green) in different populations: DDB1, OPRM1, and other variants in East Asians, and SLC24A5, SLC45A2, and HERC2/OCA2 in Europeans. The color-coded pathways—orange for high-UV eumelanin adaptation and blue for low-UV melanin reduction—highlight the role of environmental pressures in shaping pigmentation through convergent evolution, where similar phenotypes arose independently through different genetic routes, shown in green.

These findings highlight the remarkable plasticity of the human genome in responding to similar environmental pressures through distinct molecular pathways. Despite different genetic underpinnings, both evolutionary routes converged on a shared phenotype: reduced melanin content to support vitamin-D synthesis in UV-limited environments. Meanwhile, the consistent retention of dark pigmentation alleles in equatorial populations reflects the enduring importance of eumelanin in UV protection and folate preservation, maintaining a set of ancestral variants under continuous positive selection [14,25].

As summarized in Figure 2, the evolution of skin pigmentation reflects the balance between environmental demands and physiological needs, filtered through the lens of genetic adaptation. This dynamic interplay between natural selection, mutation, migration, and demographic history has produced the diverse pigmentation patterns observed in modern human populations—making skin color one of the most visible markers of our species’ evolutionary journey.

### 3.5. Comparative Evolutionary Context: Insights from Non-Human Primates

Understanding the evolution of human pigmentation requires placing it within a broader primate phylogenetic framework. Comparative genomic studies have shown that several pigmentation-related genes—such as MC1R, KITLG, TYRP1, and OCA2—are highly conserved among non-human primates, underscoring their fundamental roles in melanin production and UV protection. For example, MC1R exhibits strong purifying selection in chimpanzees, gorillas, and orangutans, with minimal functional variation across individuals. This conservation suggests that in these species, maintaining dark pigmentation has been evolutionarily advantageous, likely due to life in high-UV environments such as tropical forests [25].

In contrast, humans are unique among primates in exhibiting widespread functional variation in MC1R, particularly in populations from higher latitudes where lighter skin evolved. The relaxation of constraint on MC1R in modern humans has been interpreted as a response to reduced UV exposure and the corresponding need for increased vitamin-D synthesis. This shift allowed greater phenotypic diversity in pigmentation, leading to the evolution of light skin in European populations via MC1R loss-of-function variants, while East Asians followed different genetic routes (e.g., OCA2, DDB1/TMEM138), exemplifying convergent evolution.

Additional primate studies provide broader insight into the evolution of pigmentation. In rhesus macaques and vervet monkeys, variation in fur and skin color has been associated with social signaling and sexual selection rather than photoprotection alone [14,27]. In mandrills, for example, facial pigmentation varies with male dominance and is regulated by hormonal cues. These findings suggest that pigmentation traits may have multifactorial adaptive functions—including thermoregulation, camouflage, and intraspecific communication—that extend beyond UV shielding.

Placing human-pigmentation evolution within this comparative framework emphasizes both the shared genetic architecture among primates and the unique environmental and social pressures that have shaped the human lineage. It also highlights the evolutionary plasticity of pigmentation traits, which may be subject to multiple selective forces depending on ecological and behavioral context. Such insights reinforce the importance of integrating comparative genomics into models of human adaptation.

### 3.6. Regional Variation and Local Adaptation in Human Pigmentation

Although global patterns of skin pigmentation broadly reflect latitude and UV radiation intensity, finer-scale regional variation highlights the complex interplay of genetic, environmental, and cultural factors shaping human pigmentation diversity. In South Asia, for example, populations exhibit a wide range of skin tones that cannot be solely explained by UV gradients. Recent genome-wide studies have revealed signatures of local adaptation involving alleles in genes such as SLC24A5, MFSD12, and DDB1, which vary significantly within the subcontinent [28,29].

In Melanesia, particularly among Solomon Islanders, a derived allele in the TYRP1 gene has been linked to naturally occurring blond hair in the absence of European admixture. This variant is virtually absent outside of Melanesia and illustrates how similar phenotypes—like light hair or skin—can evolve via different genetic mechanisms in distinct populations [30].

Inuit populations in Arctic regions provide another striking example of region-specific adaptation. Despite limited UV radiation, these groups retain relatively darker skin, likely due to their traditional diet rich in vitamin D, which alleviates the pressure for lighter skin typically seen in high-latitude populations. This highlights how cultural practices can buffer selective pressures and influence pigmentation evolution [31].

Even within Africa, a continent often treated monolithically in pigmentation research, there is substantial intraregional variation. Studies in East-African populations, including the Maasai and the Amhara, have shown distinct allele frequencies in pigmentation-related genes, influenced by both ancient population structure and more recent selection [1].

These examples collectively underscore that pigmentation evolution is not uniform but instead reflects regionally specific histories of selection, migration, admixture, and ecological pressures. Future research must move beyond continental-scale generalizations to better understand the nuanced genetic architecture and evolutionary context of pigmentation in diverse human populations.

## 4. Integration of Multidisciplinary Approaches

The study of skin pigmentation has evolved into a truly multidisciplinary field, integrating insights from genetics, archaeology, climatology, medicine, cultural studies, and more. This comprehensive approach is essential to understanding not only the evolutionary origins of pigmentation but also its contemporary significance.

### 4.1. Ancient DNA and Evolutionary Insights

One of the most transformative advances has come from ancient DNA research. By sequencing genetic material preserved in archaeological remains, scientists can track the emergence and spread of pigmentation alleles over time and geography. These genetic timelines are often combined with environmental data—such as UV radiation and climate models developed by ecologists and geographers—to identify the selective pressures shaping skin-color adaptations. This integration allows researchers to reconstruct how human populations adapted to diverse environments across millennia.

### 4.2. Environmental and Climatic Context

Mapping allele frequencies onto environmental variables such as solar radiation intensity, altitude, and latitude enables a nuanced understanding of how natural selection acted on pigmentation genes [32]. These ecological models reveal correlations between lighter skin pigmentation and lower UV environments, supporting hypotheses about vitamin-D synthesis as a key driver of evolutionary change.

### 4.3. Cultural and Social Dimensions

Anthropologists and historians provide crucial perspectives on the cultural meanings of skin color. Their work explores how pigmentation has been interpreted and politicized in various societies, shaping identity, social status, discrimination, and beauty standards. This socio-cultural context enriches our understanding of pigmentation beyond biology, emphasizing its role in human experience and social dynamics.

### 4.4. Molecular and Computational Advances

At the molecular level, collaborations with biophysicists have illuminated how melanin interacts with light, deepening insights into its photoprotective functions. Meanwhile, computational biologists develop sophisticated models simulating gene networks involved in melanogenesis, revealing complex regulatory mechanisms underlying pigmentation variation.

## 5. Biomedical Implications

The evolutionary history of skin pigmentation continues to influence health outcomes in modern populations, especially as humans migrate and settle in environments different from those of their ancestors.

### 5.1. UV Exposure and Skin-Cancer Risk

Individuals with lighter skin living in high-UV regions face increased risks of skin cancer due to reduced melanin protection [33]. Their skin, adapted to low-UV environments, is more vulnerable to ultraviolet damage, underscoring the importance of evolutionary context in understanding susceptibility. Meta-analyses confirm heightened melanoma risk among individuals with MC1R variants, particularly in Australia and Mediterranean populations [34,35].

### 5.2. Vitamin-D Deficiency in Darker-Skinned Populations

Conversely, people with darker skin living in northern latitudes or low-UV environments may experience vitamin-D deficiency [21,33]. Their high melanin content, while protective against UV damage, limits vitamin-D synthesis under reduced sunlight, potentially leading to bone disorders and immune-system challenges.

### 5.3. Pigmentation Disorders and Genetic Background

Certain pigmentation-related disorders, such as vitiligo, melasma, and post-inflammatory hyperpigmentation, are influenced by genetic variants that once conferred adaptive advantages. Modern environmental and lifestyle factors may exacerbate these conditions, highlighting complex interactions between genetics and environment.

### 5.4. Clinical Challenges and Healthcare Disparities

Pigmentation differences can impact disease diagnosis and treatment. Medical education often relies on examples centered on lighter skin, which can hinder the recognition of conditions in darker-skinned patients and can contribute to healthcare disparities. Greater inclusivity and awareness of phenotypic diversity in clinical training are essential for equitable healthcare.

## 6. Dermatologic Conditions Related to Pigmentation

Pigmentary disorders manifest as alterations in melanin production, distribution, or melanocyte integrity. There is a delicate balance between genetic programming, cellular migration, environmental modulation, and immune surveillance in maintaining pigmentation homeostasis. The key points of this section are outlined in Table 2.

### 6.1. Vitiligo

Vitiligo is a chronic, acquired depigmenting disorder characterized by the selective and progressive loss of functional epidermal melanocytes, resulting in well-demarcated, achromic macules and patches. The pathogenesis is multifactorial, involving genetic susceptibility loci, autoimmune mechanisms, oxidative stress, and intrinsic melanocyte defects. Genome-wide association studies have identified susceptibility variants in pigmentation genes such as TYR (aka tyrosinase), OCA2, and MC1R, as well as immune-regulatory genes including HLA, NLRP1, and PTPN22, which link pigmentary and immune pathways [36].

At the cellular level, cytotoxic CD8+ T lymphocytes specifically target melanocyte antigens, mediated by dysregulated antigen presentation and cytokine production (e.g., IFN-γ, TNF-α) [37]. Oxidative stress within the melanocyte microenvironment further exacerbates cell damage, while impaired melanocyte stem-cell reservoirs compromise repigmentation capacity. Epigenetic modifications and neurochemical factors, such as neuropeptides and catecholamines, may modulate melanocyte vulnerability.

Clinically, vitiligo often presents symmetrically but may display segmental or focal patterns. It is associated with other autoimmune disorders like autoimmune thyroiditis, type 1 diabetes mellitus, and pernicious anemia, reflecting a shared autoimmune predisposition [38]. The psychosocial impact of visible depigmentation is significant, influencing patient quality of life and prompting advances in immunomodulatory therapies.

### 6.2. Melasma

Melasma is a common acquired hypermelanosis characterized by bilateral, irregular, hyperpigmented macules and patches predominantly localized to sun-exposed facial regions, including the malar cheeks, forehead, and upper lip [39]. Melasma primarily affects women, particularly during hormonal fluxes such as pregnancy (“chloasma”) or with exogenous estrogen/progestin exposure (e.g., oral contraceptives) [40].

The pathophysiology involves interactions between UV radiation, hormonal influences, and genetic predispositions. UV exposure induces reactive oxygen species, stimulating melanogenesis via the upregulation of melanogenic enzymes, notably tyrosinase and its cofactors MITF (microphthalmia-associated transcription factor) and endothelin-1, which enhances melanocyte proliferation and melanin production [41]. Hormonal factors modulate melanocyte receptor signaling, increasing melanogenesis through estrogen and progesterone receptors expressed on melanocytes.

Histologically, melasma exhibits increased melanin deposition in both epidermal and dermal layers, accompanied by vascular proliferation and basement membrane alterations [42]. Genetic studies suggest polymorphisms in pigmentation-related genes may confer increased susceptibility.

Therapeutic interventions focus on photoprotection, topical agents (e.g., hydroquinone, tretinoin, corticosteroids), and procedural modalities (chemical peels, laser therapy) [43]. Melasma exemplifies how environmental and endocrine factors converge on melanocyte function to alter pigmentation patterns, reflecting adaptive and maladaptive responses.

### 6.3. Post-Inflammatory Hyperpigmentation

Post-inflammatory hyperpigmentation (PIH) is a reactive hypermelanosis arising after cutaneous injury or inflammation, such as acne vulgaris, eczema, psoriasis, or physical trauma. PIH is especially prevalent in individuals with higher Fitzpatrick phototypes (IV–VI), who have a greater baseline melanogenic response.

Inflammatory mediators, including prostaglandins, leukotrienes, and a spectrum of cytokines (IL-1, IL-6, TNF-α), stimulate melanocytes to increase melanin synthesis via activation of tyrosinase and other melanogenic enzymes [44]. Additionally, inflammatory damage to keratinocytes disrupts melanosome transfer regulation, facilitating abnormal pigment dispersion and retention in the basal epidermis and upper dermis.

Clinically, PIH manifests as localized brown-to-gray macules or patches that correspond to the site of prior inflammation. The pigmentation may persist for months to years, depending on injury severity and individual repair mechanisms. PIH highlights melanocytes’ sensitivity to inflammatory microenvironments and the dynamic regulation of pigment production in response to tissue damage, underscoring the need for anti-inflammatory management to prevent or mitigate hyperpigmentation sequelae [45].

### 6.4. Piebaldism

Piebaldism is a rare congenital pigmentary disorder inherited in an autosomal dominant fashion, characterized by stable, non-progressive depigmented patches of skin and a distinctive white forelock. The underlying molecular defect involves loss-of-function mutations in the KIT proto-oncogene receptor tyrosine kinase, which plays a pivotal role in melanoblast proliferation, migration, and survival during embryogenesis [46].

Defective KIT signaling impairs melanocyte colonization of the skin and hair follicles, resulting in congenital absence of melanocytes in affected areas. The patches are typically bilateral and symmetrical, mostly located on the anterior trunk, forehead, and extremities. Unlike acquired pigmentary disorders, piebaldism reflects developmental melanocyte defects rather than melanocyte destruction. The white forelock is pathognomonic, and although pigmentation elsewhere may be normal, affected skin shows complete amelanosis without repigmentation potential.

Piebaldism illustrates the crucial role of KIT signaling in neural crest-derived melanocyte lineage establishment and highlights developmental genetics’ contribution to pigmentary phenotypes [46].

### 6.5. Waardenburg Syndrome

Waardenburg syndrome is part of a group of genetic disorders characterized by congenital pigmentary anomalies, sensorineural hearing loss, and craniofacial dysmorphisms. It results from mutations in neural crest regulatory genes, including PAX3 (type I and III), MITF (type II), SOX10, EDNRB, and EDN3, which orchestrate melanoblast specification, migration, differentiation, and survival [46,47].

Pigmentary manifestations include heterochromia iridis (different colored eyes), white forelock, and patchy hypopigmentation of the skin and hair. Sensorineural deafness occurs due to melanocyte loss in the stria vascularis of the cochlea. Clinical subtypes vary in severity and associated anomalies, such as dystopia canthorum (lateral displacement of inner eye corners) and limb abnormalities. Waardenburg syndrome exemplifies the developmental dependency of melanocyte lineages on neural crest pathways and their pleiotropic effects on multiple organ systems.

### 6.6. Oculocutaneous Albinism (OCA)

Oculocutaneous albinism constitutes a group of genetically heterogeneous autosomal recessive disorders defined by partial or complete absence of melanin in the skin, hair, and eyes. Defects occur in key melanogenic enzymes or transporters, including tyrosinase (TYR, OCA1), P protein (OCA2), tyrosinase-related protein 1 (TYRP1, OCA3), and SLC45A2 (OCA4), disrupting the biosynthesis and distribution of eumelanin and pheomelanin [48].

Clinically, OCA is characterized by hypopigmentation ranging from white-to-light-yellow hair, very pale skin, and characteristic ocular abnormalities including nystagmus, reduced visual acuity, foveal hypoplasia, and photophobia due to defective melanin in the retinal pigment epithelium and iris. The lack of protective melanin increases vulnerability to UV-induced DNA damage and therefore skin-cancer risk.

OCA shows the essential enzymatic and molecular machinery required for melanin synthesis, and its study has advanced understanding of melanogenesis, melanosome biogenesis, and pigment cell biology. Furthermore, OCA phenotypes illustrate evolutionary trade-offs since mutations impairing pigmentation confer clinical vulnerability but persist due to recessive inheritance and population genetic factors.

## 7. Discussion

### 7.1. Genetic Architecture and Evolutionary Adaptation

The genetic basis of human skin pigmentation is complex, involving multiple genes with varying effects that have been shaped by natural selection to optimize survival in diverse UV environments. Key pigmentation genes such as MC1R, SLC24A5, TYR, and OCA2 illustrate how different allelic variants confer adaptive advantages depending on geographical UV exposure [49,50]. Darker pigmentation alleles, which enhance eumelanin production, have been strongly conserved in equatorial populations due to their role in protecting against UV-induced DNA damage and preserving folate, a critical nutrient for reproductive success.

Conversely, the emergence of lighter skin pigmentation in populations inhabiting high-latitude, low-UV regions represents a classic example of convergent evolution. Europeans and East Asians developed lighter skin independently via distinct genetic variants, highlighting that similar environmental pressures can drive parallel phenotypic outcomes through different molecular pathways. This divergence underscores the evolutionary plasticity of the human genome, revealing multiple genetic routes to similar adaptive ends.

### 7.2. Epistatic and Epigenetic Regulation

Beyond individual gene effects, pigmentation phenotypes arise from complex gene–gene interactions and epigenetic modifications. Epistatic relationships, such as the antagonism of MC1R by ASIP, modulate melanin-synthesis pathways and contribute to the diversity of pigmentation patterns [51,52]. These interactions mean that the phenotypic impact of a single gene variant can be amplified or suppressed depending on the presence of other alleles, complicating the mapping of genotype to phenotype.

Epigenetic mechanisms add another regulatory layer, allowing pigmentation genes to respond dynamically to environmental factors, especially UV radiation. UV-induced changes in DNA methylation and histone modification patterns within melanocytes influence gene expression without altering the DNA sequence, providing phenotypic plasticity that can adaptively fine-tune pigmentation in response to fluctuating environmental conditions. This regulatory flexibility may contribute to rapid acclimatization beyond slower genetic adaptation.

### 7.3. Multidisciplinary Insights into Pigmentation Evolution

Recent advances in ancient DNA sequencing, coupled with environmental and archaeological data, have revolutionized the understanding of pigmentation evolution [33,53]. Ancient genomes reveal the temporal emergence and spatial spread of pigmentation alleles, allowing the reconstruction of selective pressures and migration patterns that shaped current phenotypic distributions. Ecological models integrating solar radiation intensity, altitude, and climatic variables support hypotheses linking pigmentation variation to vitamin-D synthesis and photoprotection.

Anthropological and cultural perspectives enrich this biological narrative by exploring the socio-cultural meanings attached to skin color throughout human history. Skin pigmentation has influenced social identity, beauty standards, and systems of discrimination, which in turn may have shaped selective regimes through mate choice or social structuring. These socio-cultural dimensions emphasize that pigmentation is not merely a biological trait but a key element of human experience and society.

### 7.4. Biomedical and Public Health Implications

The evolutionary legacy of pigmentation has direct relevance to contemporary health disparities. Individuals with lighter skin in high-UV environments face increased skin-cancer risk due to insufficient melanin protection, while darker-skinned individuals in low-UV regions often experience vitamin-D deficiency, posing risks for bone health and immune function. Understanding these evolutionary mismatches is critical for developing targeted public-health interventions and personalized medical care.

Moreover, pigmentation-related genetic variants influence susceptibility to disorders such as vitiligo, melasma, and post-inflammatory hyperpigmentation. These conditions exemplify the complex interplay between genetics, environment, and lifestyle in health outcomes. Additionally, medical education often underrepresents pigmentation diversity, leading to diagnostic challenges and healthcare inequities, underscoring the need for greater inclusivity in clinical practice. The points made in this section are outlined in Table 3.

## 8. Conclusions

Human skin pigmentation exemplifies the intricate interaction between genetics, environment, and culture that shapes human diversity. Far from a superficial trait, pigmentation reflects deep evolutionary adaptations to UV radiation, balancing protection against DNA damage with the physiological need for vitamin-D synthesis. The convergent evolution of lighter skin pigmentation in distinct populations demonstrates the versatility of human genomes in responding to similar environmental pressures through different genetic pathways.

This review highlights that pigmentation is governed by complex genetic networks and regulatory mechanisms, influenced by both historical selective pressures and ongoing environmental interactions. The integration of genetic, ecological, archaeological, and anthropological research offers a comprehensive understanding of pigmentation as an adaptive trait and a marker of human evolutionary history.

The biomedical consequences of pigmentation variation underscore the continuing impact of our evolutionary past on present-day health. Recognizing the diversity of pigmentation phenotypes is crucial for equitable healthcare, disease prevention, and medical research.

Ultimately, the study of human skin pigmentation serves as a powerful example of how visible traits can illuminate broader principles of adaptation, biology, and identity—reminding us that our diversity is a testament to the resilience and complexity of our species.

## 9. Future Directions

Future research on human skin pigmentation should prioritize expanding genomic studies to include a wider diversity of global populations, particularly underrepresented groups, to uncover novel variants and better understand polygenic influences across environments. Integrative multi-omics approaches combining genomics, transcriptomics, and epigenomics in melanocytes and related cells will be essential to reveal the complex regulatory networks that govern pigmentation and how these are modulated by environmental factors like UV exposure. Longitudinal studies tracking gene-environment interactions over time can provide insight into the dynamic plasticity of pigmentation traits and their health consequences. The functional characterization of genetic variants through experimental models will clarify their biological roles and support targeted therapies for pigmentary disorders. Additionally, interdisciplinary work incorporating sociocultural and evolutionary perspectives is needed to explore how cultural practices and sexual selection have shaped pigmentation diversity. Finally, translating these insights into clinical and public-health contexts by developing personalized risk assessments and enhancing medical education on pigmentation diversity will be crucial to addressing health disparities related to skin cancer, vitamin-D deficiency, and pigmentation disorders.

## Figures and Tables

**Figure 1 biology-14-01026-f001:**
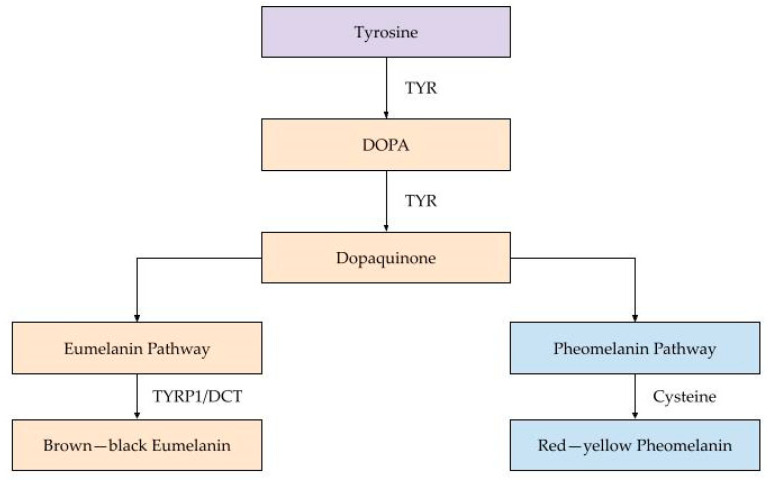
Melanogenesis pathway in human skin pigmentation.

**Figure 2 biology-14-01026-f002:**
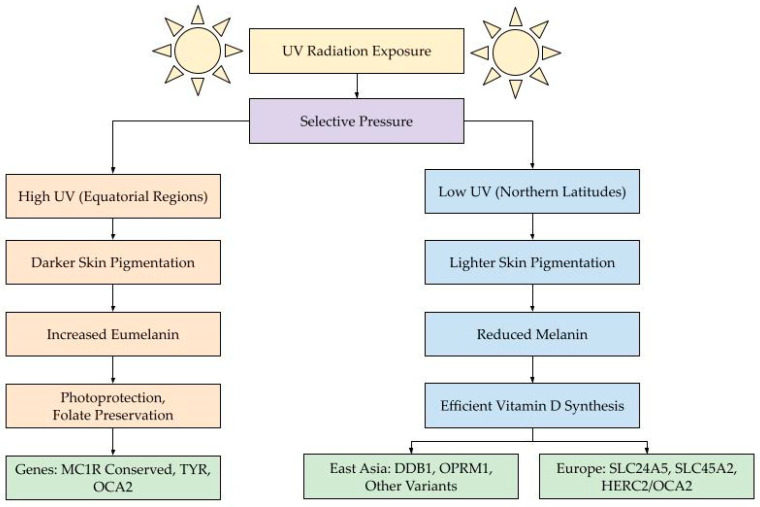
Conceptual diagram of skin pigmentation evolution in response to UV radiation.

**Table 1 biology-14-01026-t001:** Key genes involved in human skin pigmentation and their functions.

Gene	Protein Function	Effect on Pigmentation	Associated Variants	Population Prevalence
MC1R	Receptor for melanocortins	Regulates eumelanin vs. pheomelanin production	R151C, R160W, D294H	Common in Europeans
SLC24A5	Cation exchanger in melanosomes	Affects melanosome pH and melanin synthesis	A111T	High in Europeans, low in Africans
TYR	Tyrosinase enzyme, key in melanin biosynthesis	Rate-limiting step in melanogenesis	R402Q	Global, variable frequency
OCA2	Melanosomal transporter	Influences melanosome pH, eye/hair/skin color	rs1800407 (Arg419Gln)	Common in Europeans and Asians
SLC45A2	Membrane transporter involved in melanin synthesis	Affects pigmentation level	L374F	Common in Europeans

**Table 2 biology-14-01026-t002:** Summary of dermatologic conditions related to pigmentation.

Disorder	Cause/Genes	Pathophysiology	Clinical Features	Significance
Vitiligo	Autoimmune; TYR, OCA2, MC1R	CD8+ T-cell melanocyte destruction	Depigmented patches; autoimmune	Immune-mediated melanocyte loss
Melasma	Hormonal, UV exposure	Increased tyrosinase, MITF activity	Symmetric facial hyperpigmentation	Hormonal and environmental effects
Post-Inflammatory Hyperpigmentation	Inflammation	Cytokines increase tyrosinase activity	Hyperpigmentation post-injury	Melanocyte response to inflammation
Piebaldism	KIT mutation (dominant)	Impaired melanoblast migration	Congenital depigmented patches, white forelock	Defective embryonic melanocyte migration
Waardenburg Syndrome	PAX3, MITF, SOX10, EDNRB mutations	Neural crest melanocyte development defects	Heterochromia, white forelock, deafness	Neural crest melanocyte lineage disorder
Oculocutaneous Albinism	Autosomal recessive TYR, OCA2, TYRP1, SLC45A2	Defective melanin synthesis enzymes	Hypopigmentation, ocular problems	Essential melanogenesis, UV protection

**Table 3 biology-14-01026-t003:** Overview of key themes in the genetic and evolutionary study of human skin pigmentation.

Theme	Summary	Implications
Genetic Adaptation	Multiple pigmentation genes have undergone natural selection in response to UV radiation. Convergent evolution of light skin occurred via different genetic pathways in Europe and East Asia.	Demonstrates the adaptive flexibility and genetic diversity underlying human pigmentation.
Gene Interactions and Epigenetics	Epistatic interactions (e.g., *MC1R–ASIP*) and environmentally responsive epigenetic modifications influence pigmentation phenotypes.	Highlights complexity in linking genotype to phenotype and the role of regulatory mechanisms.
Multidisciplinary Insights	Integration of ancient DNA, climatic models, and anthropological data provides a richer understanding of pigmentation evolution.	Emphasizes the value of interdisciplinary approaches in evolutionary genetics.
Biomedical and Public Health Implications	Skin pigmentation affects disease risk (e.g., skin cancer, vitamin-D deficiency) and clinical outcomes due to bias in medical education and diagnostics.	Supports development of equitable healthcare and personalized medicine strategies.
Future Directions	Further research is needed in underrepresented populations, epigenetic regulation, and socio-environmental influences.	Advocates for inclusive, holistic studies to advance understanding and improve health equity.

## Data Availability

No new data were created or analyzed in this study.

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
