# Peer review of "The Genetics and Evolution of Human Pigmentation"

_biology, 2025, doi:10.3390/biology14081026_

Round 1
Reviewer 1 Report
Comments and Suggestions for Authors
The manuscript provides a review of the genetics and evolution of human pigmentation. I thought the paper was well-structured . There is some repetition, but that doesn't detract from the presentation of the material.
My only (minor) comment is that the affiliation of the second author, Elie Saliba, is not given. Of course, there may be a good reason for this.
Author Response
Reviewer 1:
Comment 1: The manuscript provides a review of the genetics and evolution of human pigmentation. I thought the paper was well-structured . There is some repetition, but that doesn't detract from the presentation of the material.
Response 1: Thank you for your valuable feedback. In response to your comment, we have minimized repetition throughout the manuscript and consolidated sections with overlapping content. Specifically, Sections 3.4 and 3.5 have been merged into a single section, and the future directions sections have also been combined for improved clarity and coherence.
Comment 2: My only (minor) comment is that the affiliation of the second author, Elie Saliba, is not given. Of course, there may be a good reason for this.
Response 2: We have made sure there is an affiliation added for Dr. Saliba.
Reviewer 2 Report
Comments and Suggestions for Authors
This is very superficial review. The text is not very long, which wouldn’t be a problem, it could be a mini review. However, what worries me the most is number of citations - 21. Many sections have no citations at all, e.g. Section 3.4 and 3.5, and others.
Figures are not sufficiently described. For instance in Figure 1 boxes are of different colors but the meaning of the colors is not explained.
Section 3.4 and 3.5 have the same titles and they are somehow redundant. I suggest to merge them and make them more consistent.
In conclusion, although this review touches very interesting topic its overall value is very limited. Consequently, I DO NOT recommend it for publication in Biology.
Author Response
Reviewer 2:
Comment 1: This is very superficial review. The text is not very long, which wouldn’t be a problem, it could be a mini review. However, what worries me the most is number of citations - 21. Many sections have no citations at all, e.g. Section 3.4 and 3.5, and others.
Response 1: Thank you for your valuable feedback. In response to Reviewer 1's comments, we have incorporated several new sections of text to address the concerns raised. Additionally, we have included citations not only in the sections mentioned but also in various other relevant parts of the manuscript. This has increased the total number of citations to 53.
Comment 2: Figures are not sufficiently described. For instance in Figure 1 boxes are of different colors but the meaning of the colors is not explained.
Response 2: We sincerely appreciate your insightful comments. In response, we have provided detailed and thorough captions for all figures to enhance clarity and ensure the content is more comprehensible.
Comment 3: Section 3.4 and 3.5 have the same titles and they are somehow redundant. I suggest to merge them and make them more consistent.
Response 3: We appreciate your valuable feedback. In response to your comments, we have merged the sections into a single Section 3.4 and revised the content to reduce redundancy. Thank you for helping us improve the clarity and coherence of our manuscript.
Reviewer 3 Report
Comments and Suggestions for Authors
Manuscript “The Genetics and Evolution of Human Pigmentation” by Guermazi and Saliba is well written, providing a comprehensive overview of skin pigmentation, from basic melanin biology to complex genetic interactions, evolutionary pressures, and clinical relevance. The writing is clear, concise, and effectively conveys complex scientific concepts.
Specific observations and suggestions:
- Sections 3.4. and 3.5. are very similar in content (but also of the same name), merge them into one.
- There are two "Future Directions" sections: one at the end of the "Discussion" (6.5) and a separate section (8). Combine them into one, comprehensive “Future Directions” section.
- Line 248: Perhaps you could briefly mention some examples, such as UV exposure for vitiligo or melasma.
Sincerely
Author Response
Comment 1: Manuscript “The Genetics and Evolution of Human Pigmentation” by Guermazi and Saliba is well written, providing a comprehensive overview of skin pigmentation, from basic melanin biology to complex genetic interactions, evolutionary pressures, and clinical relevance. The writing is clear, concise, and effectively conveys complex scientific concepts.
Response 1: We thank the reviewer for their positive feedback.
Comment 2: Sections 3.4. and 3.5. are very similar in content (but also of the same name), merge them into one.
Response 2: We appreciate your valuable feedback. In response to your comments, we have merged the sections into a single Section 3.4 and revised the content to reduce redundancy. Thank you for helping us improve the clarity and coherence of our manuscript.
Comment 3: There are two "Future Directions" sections: one at the end of the "Discussion" (6.5) and a separate section (8). Combine them into one, comprehensive “Future Directions” section.
Response 3: Thank you for your valuable feedback. In response to your comment, we have removed the text from Section 7 regarding future directions and retained only the current content in Section 9 under “Future Directions.” We believe this revision enhances the clarity and focus of the manuscript.
Comment 4: Line 248: Perhaps you could briefly mention some examples, such as UV exposure for vitiligo or melasma.
Response 4: Thank you for your valuable feedback. We kindly direct your attention to Section 6.2, which specifically addresses Melasma. Additionally, we have condensed the section containing this line, and we hope that the revisions meet your expectations. Please let us know if further adjustments are needed.
Round 2
Reviewer 2 Report
Comments and Suggestions for Authors
I welcome much improved manuscript. I think it can be published in the current form.